# Rising tension in the Himalayas: A geospatial analysis of Chinese border incursions into India

Jan-Tino Brethouwer[1], Robbert Fokkink[1]*, Kevin Greene[2], Roy Lindelauf[3], Caroline Tornquist[4], V. S. Subrahmanian[5]

1 Delft University of Technology, Institute of Applied Mathematics, Delft, Netherlands, 2 Princeton University, School of Public and International Affairs, Princeton, NJ, United States of America, 3 Netherlands Defence Academy, Faculty of Military Sciences, Breda, Netherlands, 4 Dartmouth College, Department of Computer Science, Hanover, NH, United States of America, 5 Department of Computer Science and Buffett Institute for Global Affairs, Northwestern University, Evanston, IL, United States of America

* r.j.fokkink@tudelft.nl

**Data Availability Statement:** All relevant data are within the paper and its Supporting information files.

## Abstract

The China-India border is the longest disputed border in the world. The countries went to war in 1962 and there have been recurring border skirmishes ever since. Reports of Chinese incursions into Indian territory are now a frequent occurrence. This rising tension between the world's most populous countries not only poses risks for global security and the world economy, but also has a negative impact on the unique ecology of the Himalayas, because of the expanding military infrastructure. We have assembled a unique data set of the dates and locations of the major incursions over the past 15 years. We find that the conflict can be separated into two independent conflicts, the western and eastern sectors. The incursions in these sectors are statistically independent. However, major incidents do lead to an increased tension that persists for years all along the entire Line of Actual Control (LAC). This leads us to conclude that an agreement on the exact location of a limited number of contested regions, such as the Doklam plateau on the China-Bhutan border, has the potential to significantly defuse the conflict, and could potentially settle the dispute at a further date. Building on insights from game theory, we find that the Chinese incursions in the west are strategically planned and may aim for a more permanent control over specific contested areas. This finding is in agreement with other studies into the expansionist strategy of the current Chinese government.

## Introduction

On 15 June 2020, an Indian patrol and a Chinese patrol clashed in the Galwan river valley, a disputed area along the China-India border. The patrols were unarmed in accordance with bilateral agreements, but the soldiers used rocks and sticks. This resulted in the deaths of 20 Indian soldiers and an unknown number of Chinese soldiers [1] (the Chinese government reported 4 deaths, while Time magazine [2] estimated 35). This was by far the most violent

**Funding:** The authors received no specific funding for this work.

**Competing interests:** The authors have declared that no competing interests exist.

border incident in years, but the number of Chinese incursions along the China-India border had been steadily increasing, and it seemed like an accident waiting to happen [3]. Immediately after the clash both sides sought to reduce the tension through rounds of talks [4], meanwhile increasing their military presence [5]. In 2021 they agreed to withdraw their troops from a number of red-zones and the number of incursions now seems to be dropping [6].

The Line of Actual Control (LAC) between China and India remains the longest disputed land border on earth. After a brief war in 1962, the two countries signed several bilateral agreements. In 2005, this culminated in a protocol [7] to develop a long-term constructive partnership, pending an ultimate resolution of the conflict. The countries agreed to not use force, nor threaten to use force, against each other. However, since then, the relationship has deteriorated for several reasons. The trade surplus that India enjoyed up until 2005 has turned into a multibillion dollar deficit, which has created strong undercurrents of mistrust [8, 9]. Moreover, China significantly increased both its military spending and its military support for Pakistan, which raised a high degree of concern in India [10].

China and India possess nuclear weapons, but both countries subscribe to a no-first-use policy, and the risk of nuclear escalation seems minimal. However, the border conflict could escalate into the use of conventional missiles. Experts have pointed out that these are co-located with nuclear missiles, and that it is challenging to distinguish between the two [11]. The risk of nuclear escalation therefore is minimal, but not negligible. Meanwhile, the conventional escalation of the conflict is ongoing. Tens of thousands of troops are now stationed across the mountains [12]. To support their military presence, both countries keep extending their infrastructure, which leads to ecological degradation of the borderlands [13].

This paper seeks to build an understanding of what drives the conflict. We have assembled an original data set on Chinese incursions into India over the period 2006–2020, starting at a time of detente and ending with a year of heightened tension, when the pandemic had not yet disrupted the world. We do not consider border incursions by India into China because there are relatively few reports of such incursions during the 15 year period of our study. We apply statistical and game-theoretic methods to study the dynamic of rising and falling tensions along the LAC. Our findings provide a glimmer of hope for a possible way out of the conflict through a step-by-step de-escalation. A resolution of the conflict would be of great benefit for international security, the world economy as well as the economies of the two countries (which are strongly linked), and the ecological preservation of the Himalayas.

## Status quo of the LAC

The LAC separates the territories controlled by China and India and is perceived by both sides as a kind of working border. It was first suggested by Chinese prime minister Zhou Enlai in a letter to Indian prime minister Jawaharlal Nehru in 1959 [14]. It is a legacy of agreements between foreign powers. Indeed, a part of it is still named after a British administrator. Historically, the LAC is divided into three sectors [14–16], as illustrated in Fig 1:

1. The western sector from the Karakoram pass to Mount Gya, along Chinese controlled Aksai Chin. The Galwan river valley is located here.

2. The middle sector from Mount Gya to the border with Nepal, which is the least contested part of the border.

3. The eastern sector, also known as the McMahon line, along the state of Arunachal Pradesh on the Indian side. This sector also includes the border between Sikkim and Tibet.

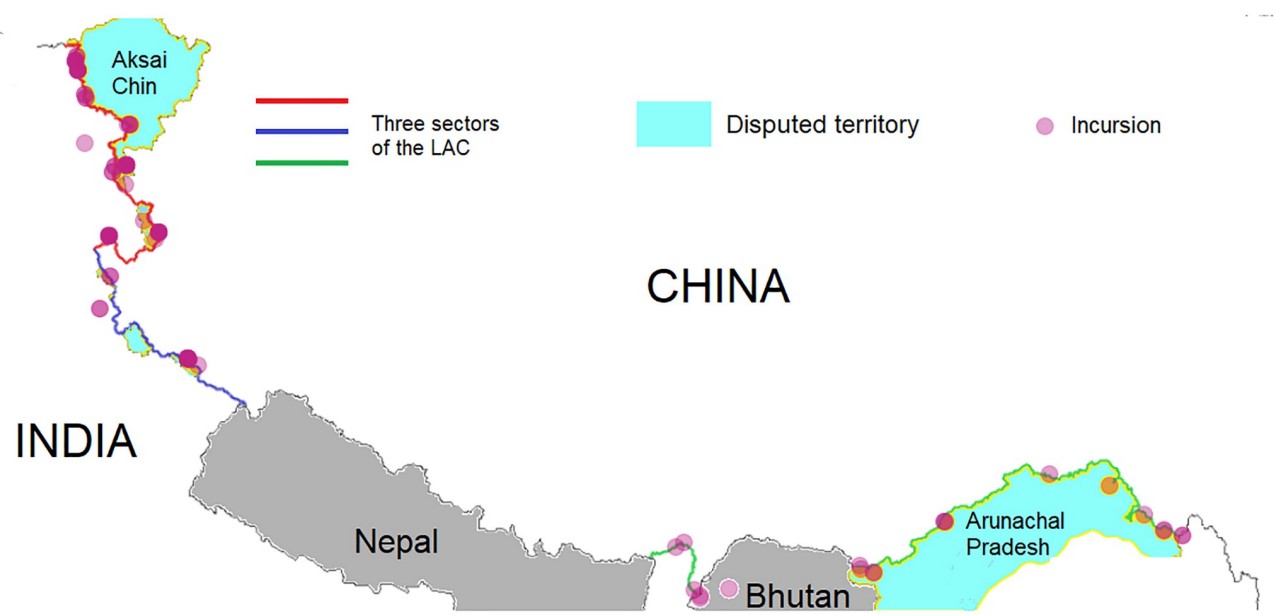

**Fig 1. A map of the LAC and its three sectors.** The locations of incursions along the LAC. The shade represents the number of incursions. The largest contested territories are Aksai Chin in the west and Arunachal Pradesh in the east. (Topographic base maps in our article are accessed from the USGS National Map Viewer).

Formally, both countries claim large areas that are controlled by the other side. The Indian claim of Chinese controlled Aksai Chin, a highland of 40,000 square kilometers, was the major cause of the short war in 1962 [15]. This virtually uninhabited plateau is a vital passage point between the Chinese autonomous regions of Tibet and Xinjiang [17]. China in turn claims 80,000 square kilometers of Arunachal Pradesh, and briefly occupied it during the 1962 war. Some speculate that China will accept the McMahon line as its border, if India waives or sharply modifies its claim to Aksai Chin [18, 19]. Other analysts believe that India wishes to freeze the status quo, a position for which China has little respect [20, 21].

## Geographical, political, and strategic aspects of the conflict

Territorial conflicts usually involve ethnic, geographic or economic conditions [22]. We survey several factors that may play a role in the conflict.

**Geographical factors.** The LAC is exceptionally long and the conditions along the border vary considerably. The borderland along the middle and eastern sectors is more accessible, fertile and densely populated than the Aksai Chin in the western sector, which is a barren plateau with a total population of less than 10,000. It is, however, of great economic importance to China, since Aksai Chin connects Tibet and Xinjiang. It is near the China-Pakistan Economic Corridor, a major project to connect Xinjiang to Pakistan's Gwadar port [23].

In the eastern sector, the Siliguri corridor is 130 km away from the Sikkim border. This narrow stretch of land is the only connection between India's North-Eastern states and the rest of the country. The Doklam standoff in 2017 was caused by intended Chinese road constructions that were perceived by India as a threat to this corridor [24].

The borderland is rich in mineral reserves, and there are several mines that are exploited or explored. The Aksai Chin has one of the largest zinc-lead deposits in the world, which is currently being prepared for exploitation [25]. Fifty kilometers north of the McMahon line, China

launched large-scale investments into mining gold and silver in Lhünzê County. Another aspect is the future use of hydro-electric power. The McMahon line runs through the Brahmaputra river basin, where both China and India have plans for the construction of major hydropower plants [26].

**Strategic factors.**   The People's Republic Army (PLA) is a dominant force. China's policy is increasingly assertive, employing a coercive policy against Taiwan, while staying in the gray zone below open conflict [27, 28]. It is continually extending control over the South China Sea. Its dispute with Japan over the Senkaku/Diaoyudao islands shows the same rising tension as the dispute over the LAC [29].

According to Indian military strategists, the number of Chinese patrols in Indian territory is increasing each year [30]. The PLA has constructed military structures, and even entire villages, in disputed areas of the LAC [31–34]. The PLA strategy is often referred to as *salami slicing* by the Indian media. On the other hand, the terrain around the LAC is more accessible from the Indian side [12] and India's military bases are much nearer to the disputed areas than China's [24]. The balance of (military) power in the conflict is unclear.

**Political factors.**   China and India have close economic ties and would benefit from a settlement of the border dispute, but they are also the two dominant powers in the region and political rivals. The dispute goes through a dynamic between cooperation and conflict, see Table 1, that is typical for the political relations between rival powers [36]. In recent years, India's prime minister Modi and China's president Xi Jinping established close personal relations [37]. After the Doklam standoff in 2017, they sought to overcome strategic differences through several informal summits. This was disrupted by the Galwan standoff in 2020, which was possibly triggered by India's decision to dissolve the special status of the state of Jammu and Kashmir [38]. The incident has been a political setback for prime minister Modi, whose Bharatiya Janata Party (BJP) stands for India's territorial unity more strongly than any other party [39].

India's participation in the Quad [40], the security dialogue between the United States, India, Japan and Australia, may have served as a trigger for Chinese activity along the China–India border. The incursions in the Aksai Chin could be a statement from Beijing to both New Delhi and Washington, that it will continue to ensure its regional interests [14].

## Methods

### Data

We built a data-set of Chinese incursions into disputed areas from 2005–2020. We define incursions as unauthorized Chinese entries into areas that are internationally accepted as either Indian or disputed territories. We do not distinguish between foot patrols, motor patrols, or combat air patrols. We collected the data using journalistic and academic sources

**Table 1. Timeline of the border conflict—A concatenation of military confrontations and bilateral agreements.**

| | |
|---|---|
| 2005 | Protocol on confidence building measures along the LAC. |
| 2012 | Agreement on a Working Mechanism for Consultation and Coordination on India-China Border Affairs [35]. |
| 2013 | *Depsang standoff* followed by the Border Defence Agreement to "establish peace and tranquility". |
| 2014 | *Chinese incursion in Chumur* just ten days before Xi visits India and signs a twenty billion dollar investment agreement. |
| 2017 | *Doklam Plateau standoff* |
| 2018 | Informal talks between Modi and Xi |
| 2020 | *Galwan River Valley standoff* |

from the LexisUni database. It therefore is as much a study of the media attention as of the incursions. We rely on reports of Chinese incursions that are well documented by multiple independent media outlets and are reported by major Indian newspapers or international media. We do not record incursions that are reported by the Indian government but not verified by a third party (India currently ranks 150 out of 180 on the World Press Freedom Index of Reporters without Borders (RSF), China ranks 175). The focus of this study is on Chinese incursions into India. We do not consider Indian incursions into China, as these there are few instances that can be verified across multiple independent sources. Most often these incursions are reported only by the Chinese state media.

The Indian government publishes yearly numbers of border incursions and transgressions, with an average of 334 incursions per year over the period 2006–2019, see [29]. The number of incursions in 2020 has not been reported yet. The government numbers include minor incidents, such as finding cans of food consumed by Chinese patrols on Indian territory, that are not reported by the media. The official numbers are compared to our numbers in Table 2. The numbers reported in the media have a higher coefficient of variation (st.dev/mean) than the official numbers (0.70 versus 0.40). In other words, the numbers presented by the media are more volatile. It is likely that this higher volatility is a consequence of the self-reinforcing nature of media attention [41]. We should keep in mind that our data measures the number of incursions through a slightly distorted lens.

We found an average number of 8.0 incursions per year that were reported by the media (our data set). If we adjust the scales to compare the two time series, as illustrated in Fig 2, then they show the same trend. The number of incursions has been rising since 2006. There have been years in which the numbers were stable or even reduced, but they were invariably followed by a further increase.

**Table 2. Validation of our data.** The Indian government publishes yearly numbers of border incidents (numbers of the Indo Tibetan Border Police), which are compared to the numbers reported by the media in our data. The official numbers correlate well with our data. The official numbers peaked in 2014 and again in 2019. Remarkably, the number of media reports on incursions in 2019 was very low.

| year | Gov India | Our data |
|---|---|---|
| 2020 | - | 14 |
| 2019 | 663 | 3 |
| 2018 | 326 | 18 |
| 2017 | 426 | 16 |
| 2016 | 273 | 9 |
| 2015 | 290 | 5 |
| 2014 | 460 | 14 |
| 2013 | 411 | 14 |
| 2012 | 426 | 6 |
| 2011 | 213 | 6 |
| 2010 | 228 | 5 |
| 2009 | 270 | 6 |
| 2008 | 280 | 2 |
| 2007 | 140 | 1 |
| 2006 | 265 | 1 |
| mean $\mu$ | 334 | 8.0 |
| stdev $\sigma$ | 132 | 5.6 |
| $\sigma/\mu$ | 0.40 | 0.70 |

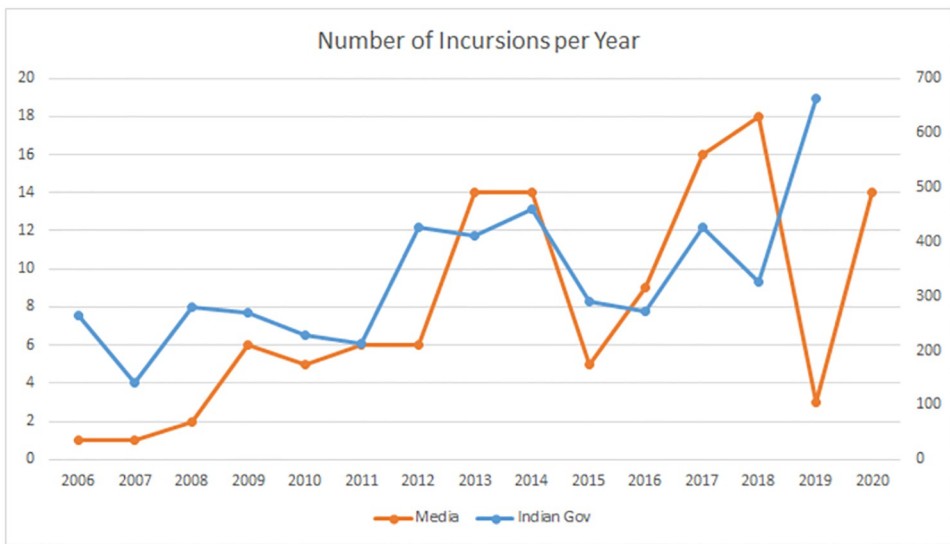

**Fig 2. Comparison of our data and the official numbers on adjusted scales.** The upward trend over the entire period is the same in both data sets, but the numbers in our data are clearly more volatile than the official numbers.

The Indian government reported no less than 663 transgressions in 2019, but only 3 of these made it to the international media. Indian prime minister Modi and Chinese president Xi held several informal meetings at the time and reiterated their effort to ensure peace, in accordance with the 2005 protocol [42]. The chief of the Indo-Tibetan Border Police (ITBP) reported a peaceful situation [43]. These efforts to downplay the actual situation, with an all-time high number of transgressions, worked: the media reported an exceptionally low number of incursions that year. If we delete this outlier from our data set, the correlation between the government numbers and our data is 0.69. All things considered, our data agrees well with the official numbers of the Indian government.

A data set that is similar to ours, though much more restricted, has been assembled by the Observer Research Foundation [44]. This data set contains only biennial numbers of incursions over the years 2006–2014. The ORF data set is very close to ours (corr. coeff. 0.99).

## Data processing

Media reports rarely specify the exact time or location of the incursions. They specify the date and the name of a village, a military base, or a geographical landmark close to where the incursion took place. It is impossible to pinpoint the incursions with absolute precision in time or space. Our data contains the date of the incursion and a best estimate for its geographic coordinates. The LAC runs across mountain crests and only a few passes and valleys can be patrolled, so that these coordinates should be approximately correct, within a few kilometers of the actual locations. A heat map of the incursions in Fig 3 clearly shows that the incursions are clustered in space.

The incursions in the middle sector are concentrated around the Barahoti military base. In the eastern sector there are six red-zones, one along the Sikkim border and five along the McMahon line. The data is more diffuse in the western sector, where the media reports incursions near six locations: Depsang, Pangong, Demchok, Chumur, Hot Springs, and Galwan Valley. The majority of the incursions occur in four locations, see Table 3 and the heat map in Fig 4.

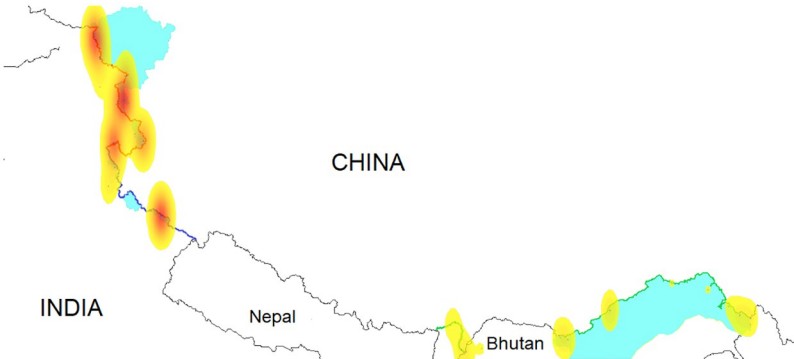

**Fig 3. A heat map of the incursions.** The media reports do not always give the exact coordinates and only mention a military base or village nearest to the incursion. It is impossible to pinpoint the precise locations of the incursions. However, it is clear that the incursions are clustered around hot spots where the border is not clearly defined. These are the so-called red-zones (USGS base map).

We treat the locations of the incursions as categorical data, divided into thirteen locations, see Fig 5. The incursions in the middle sector are all in the Barahoti area. They are more spread out in the other two sectors. In both the eastern sector and the western sector there are six red-zones.

The meteorological conditions near the LAC are forbidding with high altitudes (40 percent of the incursions occur above 5,000m) and sub-zero temperatures. As a consequence of this, the incursions are not only clustered in space, but also clustered in time, as can be seen in Fig 6. Half of the incursions (51 percent) take place during June–August, and almost none during December–February.

We remark that a detailed study of *when* incursions occur, in relation with socio-economic and political factors, was previously carried out in Greene et al. [45].

## Statistical methods

Border conflicts involve many different factors which manifest themselves on different scales. They often involve totally scattered confrontations, which can spill over from one location to the next [46, 47]. The incursions themselves are local incidents, but may be part of military strategies that are played out on a larger scale. We therefore analyze the data at different spatial and temporal levels.

The length of the LAC is enormous, with a distance of well over thousand kilometers between Indian-claimed Aksai Chin and Chinese-claimed Arunachal Pradesh. The western/middle sector and the eastern sector are separated by the independent countries of Nepal and

**Table 3. The incursions in the western and eastern sector.** The distribution is not even. In the western sector, almost all incursions occurred in Chumur, Demchok Depsang, and Pangong. In the eastern sector, three quarters of the incursions occurred in Sikkim, Tawang, and Kibithu.

| location (W) | coordinates | incursions | location (E) | coordinates | incursions |
|---|---|---|---|---|---|
| Depsang | 35.3N, 78.0E | 23% | Sikkim | 27.6N, 88.8E | 30% |
| Galwan | 34.8N, 78.2E | 6% | Tawang | 27.7N, 91.8E | 20% |
| Hotspring | 34.3N, 79.0E | 5% | Lhunze | 28.5N, 93.3E | 10% |
| Pangong | 33.7N, 79.4E | 26% | Bishing | 29.1N, 95.0E | 5% |
| Demchok | 32.8N, 79.4E | 16% | Anini | 29.0N, 96.0E | 10% |
| Chumur | 32.7N, 78.6E | 24% | Kibithu | 28.3N, 97.2E | 25% |

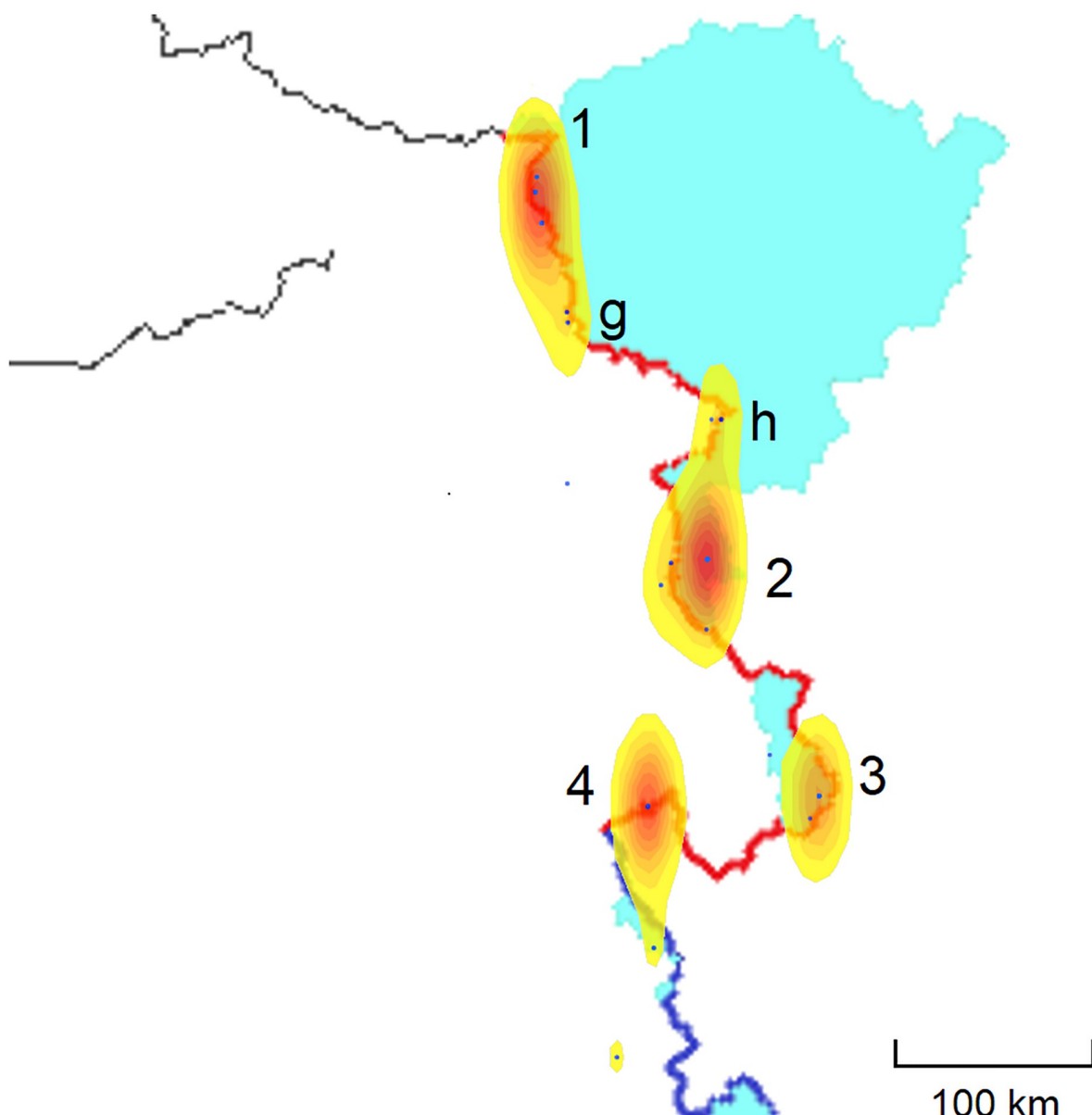

**Fig 4. Heat map of the incursions in the western sector.** The majority of the incursions occur near four red-zones: 1 Depsang Plateau, 2 Pangong Lake, 3 Demchok, 4 Chumur. Some of the incursions are outside these four red zones: near the Galwan Valley (g) or near the Hot Springs border checkpost (h), which are relatively close to each other. Pangong Lake, which is patrolled by fast boats, is 134 km long, making the incursions very difficult to locate. Chumur is close to the middle sector and the incursions here take place over a larger area, with two incursions formally located in Uttarakhand (middle sector), but close to Chumur. We group all of these into this single red-zone.

Bhutan. We test if the incursions in these two parts of the border are statistically independent events by means of a Wald-Wolfowitz run test. To this end, we label the incursions as "E" and "W" to obtain a sequence of elements for which we count the number of runs, i.e., we count the number of segments of the sequence consisting of adjacent equal elements. If the incursions in these two parts of the LAC are statistically independent, then the number of runs is asymptotically normal with mean $\mu = \frac{2n_1 n_2}{n_1 + n_2} + 1$ and variance $\sigma^2 = \frac{(\mu-1)(\mu-2)}{n_1 + n_2 - 1}$. Here $n_1$ is the

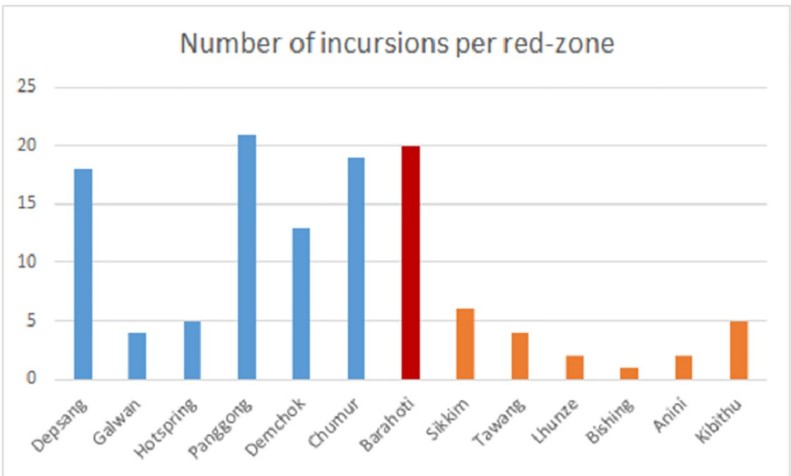

**Fig 5. Grouping the incursions into thirteen red-zones.** The distribution of the incursions over the thirteen red-zones, colored by sector (west-middle-east).

number of incursions in the East and $n_2$ is the number of incursions in the West. We use this method to test the statistical dependence of the incursions in the three sectors. We also apply a multivariate Wald-Wolfowitz test for the number of runs in the western sector. There is no closed form formula for the number of runs in a multivariate test, but the expected number of runs and its standard deviation can be simulated [48, p 217].

Mathematically, the incursions along the LAC can be modelled by a point process, in which each location on the border has a probability intensity of incursion. In such models it is customary to split a time series into a background process and an offspring process [49]. The offspring process contains the events that are triggered by the background process, as aftershocks of an earthquake. We therefore split the data set into primary incursions and secondary (triggered) incursions. We define an incursion as secondary if it occurs within 10 days in the same red-zone as an earlier incursion. The choice of 10 days is arbitrary, but the results are stable under a variation of plus or minus 3 days. It adjusts for the self-reinforcing nature of media attention, which is on a time-scale of days only [50].

In the Wald-Wolfowitz test we study incursions per sector. To understand the military strategy behind the Chinese incursions, the alleged salami-slicing, we study the incursions per red-zone. Game theory predicts that adversaries try to establish permanent control over a battleground (red-zone) by allocating more troops for a longer time than their opponents. Such a tactical allocation of troops and other military resources is a common phenomenon in border disputes [51, 52], which can be described by a Colonel Blotto game. This is a classical two-

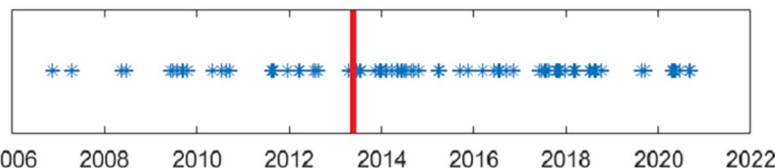

**Fig 6. The incursions through time.** The red line marks the Depsang incursion, which has been a turning point in the conflict: the number of incursions doubled since then and remained at an elevated level. The events are clearly seasonally clustered.

person constant sum game, in which the players allocate troops over several battlefields. A battlefield is won by the player that allocates more troops to that battlefield. The payoff is equal to the number of won battlefields. Of course this game is a simplification of the actual situation. Blotto is a one-shot game in which the players can access the battlefields without constraint, while in reality the border conflict is an ongoing struggle and the accessibility differs per red zone. However, analysts have pointed out that China and India are engaged in a 'war of attrition' over the red-zones, in which they try to wear each other out, while avoiding a full confrontation [10]. We interpret the Chinese incursions as attempts to establish a temporary presence in a red-zone, which can be modelled by a Blotto game that is played over rounds. We compare the yearly incursions per red-zone to optimal strategy in a Blotto game.

A pure strategy in a Blotto game is an allocation $(x_1, \ldots, x_k)$ over $k$ battlefields such that the $x_i$ add up to the total total number of troops available to the player. An optimal mixed strategy selects an allocation and then assigns the numbers $x_i$ uniformly at random to the battlefields [52]. We note that computing the specific $x_i$ is non-trivial even for a limited number of battlefields [53]. Experimental game theorists have established that humans tend to play this game more aggressively and in a more targeted manner than predicted by the mathematical optimum [54]. In particular, the player with least resources tends to follow a 'guerilla warfare' strategy, targeting a limited number of battlefields. To detect this targeting, we applied a $k$-means analysis and used silhouette scoring to determine incursion clusters on a yearly basis [55]. If a Blotto game is played over rounds, then the cumulative allocation per battlefield evens out, because it is randomly assigned. We apply a chi-square test to determine if there is a significant difference between the cumulative allocation and the average.

The number of incursions has been rising since 2005, but this has not been a steady increase. There have been several major standoffs, which seem to have sparked subsequent incursions. The Doklam standoff in 2013 was a turning point in the conflict and led to a much more volatile situation and increased media attention. To take care of this, in our time-series analysis we mainly consider primary incursions. We compute its auto-correlation to detect cycles of rising and falling tension, and we compute the correlation between east and west to check if the tension is in sync along the LAC. Finally, we apply an auto-regressive model to forecast the future development of the conflict.

## Results and discussion

### Incursions per sector

The center of gravity of the dispute is in the western sector, which has the largest number of incursions by far, see Table 4. The western sector and the middle sector are contiguous and the number of yearly incursions is weakly positively correlated (corr. coef. 0.26, 95% conf. int. (-0.28,0.81)). The eastern sector (Sikkim and the McMahon line) is disconnected from the western/middle sector, with the sovereign states of Nepal and Bhutan in between. The number of yearly incursions in the western sector and the eastern sector is uncorrelated (corr. coef. -0.11, 95% conf. int. (-0.57,0.41)). We first count the number of runs (a segment of consecutive incursions in the same sector) in the western sector versus the middle sector. It is equal to 25. The Wald-Wolfowitz run test (parameters $n_1 = 80$, $n_2 = 20$, cf. Table 4) predicts an average number of 33 runs with a standard deviation of 3.1. It rejects the hypothesis that the incursions in the western and middle sector are independent random events ($p = 0.005$). This supports the hypothesis that the western sector and the middle sector can be seen as one sector, which is in line with how China and India see it. They have described it as a single sector in their 2005 bilateral agreement.

**Table 4. Yearly incursions in three sectors of the LAC.** The incursions in the western and middle sector are weakly correlated (corr. coef. 0.26) and the western and eastern sector are uncorrelated (corr. coef. -0.11).

| year | western | middle | eastern |
|---|---|---|---|
| 2020 | 13 | 0 | 1 |
| 2019 | 1 | 0 | 2 |
| 2018 | 7 | 9 | 2 |
| 2017 | 8 | 4 | 4 |
| 2016 | 3 | 2 | 4 |
| 2015 | 4 | 0 | 1 |
| 2014 | 12 | 2 | 0 |
| 2013 | 12 | 1 | 1 |
| 2012 | 6 | 0 | 0 |
| 2011 | 5 | 0 | 1 |
| 2010 | 3 | 1 | 1 |
| 2009 | 5 | 1 | 0 |
| 2008 | 1 | 0 | 1 |
| 2007 | 0 | 0 | 1 |
| 2006 | 0 | 0 | 1 |
| total | 80 | 20 | 20 |
| mean | 5.3 | 1.3 | 1.3 |
| std.dev. | 4.4 | 2.4 | 1.2 |

If we partition the incursions into the western/middle sector versus the eastern sector, then the number of runs is 32. The Wald-Wolfowitz run test (parameters $n_1 = 100$, $n_2 = 20$) predicts 34.3 runs (st.dev 3.0), and returns a $p$-value of 0.22, which at the standard confidence level of five percent supports the hypothesis (or at least, does not refute it) that incursions in the east and west are independent random events. If we repeat this analysis for primary incursions, then the number of runs is 30 (down from 32). There are 68 primary incursions in the western/middle sector (down from 100) and 19 primary incursions in the eastern sector (down from 20). The Wald-Wolfowitz run test predicts of 30.7 runs (st. dev 2.7) and returns a $p$-value of 0.40. This supports the hypothesis that incursions in the west and in the east are independent events.

We conclude that both the correlation coefficient and the Wald-Wolfowitz test support the hypothesis that the incursions the western/middle sector and the eastern sector are statistically independent events. The border dispute can therefore be divided into two separate conflicts, centered around Aksai Chin and Arunachal Pradesh. This conclusion holds for the incursions, which are local and tactical events.

## Incursions per red-zone

In the previous section we grouped the incursions per sector. We now consider a finer scale and consider incursions per red-zone. These are areas along the border, where the exact position of the LAC is contested. Almost all incursions are in these zones. We have identified thirteen red-zones, as given in Fig 5.

The majority of these incursions are in six red-zones in the western sector. A multi-variate Wald-Wolfowitz test rejects the hypothesis that the incursions in the western sector are independent. The number of runs in our data is 50 while the expected number of runs in a Wald-Wolfowitz test is 63.3 with standard deviation 3.4 ($p = 0.0001$). This implies that the incursions are not random events and that red-zones in the western sector are strategically targeted.

**Table 5. Yearly incursions in the western sector.**

| year | Depsang | Galwan | Hotspring | Pangong | Demchok | Chumur |
|------|---------|--------|-----------|---------|---------|--------|
| 2020 | 0 | 2 | 3 | 4 | 2 | 2 |
| 2019 | 0 | 0 | 0 | 1 | 0 | 0 |
| 2018 | 6* | 0 | 0 | 0 | 1 | 0 |
| 2017 | 1 | 0 | 0 | 7* | 0 | 0 |
| 2016 | 0 | 0 | 1 | 1 | 1 | 0 |
| 2015 | 3* | 0 | 0 | 1 | 0 | 0 |
| 2014 | 1 | 1 | 0 | 3 | 2 | 5 |
| 2013 | 5 | 0 | 0 | 0 | 1 | 6* |
| 2012 | 0 | 0 | 1 | 1 | 1 | 3* |
| 2011 | 0 | 1 | 0 | 2 | 1 | 1 |
| 2010 | 1 | 0 | 0 | 1 | 1 | 0 |
| 2009 | 1 | 0 | 0 | 0 | 2 | 2 |
| 2008 | 0 | 0 | 0 | 0 | 1 | 0 |
| total | 18 | 4 | 5 | 21 | 13 | 19 |

Clusters of incursions are colored red (high), blue (middle) and green (low). During at least 75% of the years between 2012–2022 the cluster with the most incursions contains at least 3 times as much incursions as the others. Yearly clusters are determined using k-means for k = 2,3,4,5 and selecting k with highest silhouette score. The asterisk marks battlefields which receive half or more of all incursions in that year. In the years 2013, 2017, and 2018 almost all incursions targeted a single battlefield.

The number of incursions per red-zone in the western sector is given in Table 5. In at least 75% of the the last nine years (a period of heightened tension), the PLA targeted a cluster (covering at most 2 locations) containing at least 3 times as much incursions as all other incursions in the western sector. If we compare the time-line of the incursions at both ends of the western sector in Depsang and Chumur, see Fig 7, then we see that the incursions there are clustered in time. The weather is more severe at Depsang, which is at a higher altitude and on average is 5 degrees colder than Chumur. We find the same temperature difference for the dates of the incursions, which indicates that the local weather conditions do not influence the strategic allocation of the troops.

We compare the incursions in the six red-zones with optimal play in a Blotto game. Incursions are clustered using $k$-means (for $k$ with highest silhouette score) and color coded in Table 5. It demonstrates that the PLA almost always targets clusters (with 3 times the number of incursions with respect to the others) of 1 or 2 locations. In 2017 and 2018 the PLA targeted a single red-zone, and in 2013 it targeted two red-zones. If we test the maximum number of incursions in these years against a uniform distribution, then the $p$-values are respectively 0.0004, 0.002 and 0.10, which is another indication that these are not random encounters. Such a targeted strategy is consistent with an asymmetric Blotto game, in which the player with less resources assigns the troops to a limited number of battlefields and forfeits the others

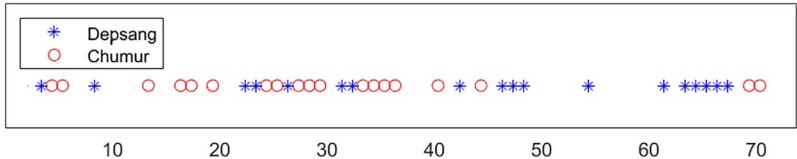

**Fig 7. Incursions in Depsang and Chumur.** The incursions in time in the two outermost points of the western sector, which are 300 km apart. The 37 incursions are numbered in following order. They are clearly clustered, indicating that one of the two zones is targeted at a time.

**Table 6. Yearly incursions in eastern sector.** The number of incursions are comparatively small. During three years there were no incursions and eight years only had one incursion.

| year | Sikkim | Tawang | Lhunze | Bishing | Anini | Kibithu |
|---|---|---|---|---|---|---|
| 2020 | 1 | 0 | 0 | 0 | 0 | 0 |
| 2019 | 0 | 0 | 0 | 0 | 0 | 2 |
| 2018 | 1 | 0 | 0 | 0 | 1 | 0 |
| 2017 | 1 | 0 | 1 | 1 | 0 | 1 |
| 2016 | 0 | 2 | 0 | 0 | 0 | 2 |
| 2015 | 0 | 0 | 0 | 0 | 1 | 0 |
| 2014* | 0 | 0 | 0 | 0 | 0 | 0 |
| 2013 | 1 | 0 | 0 | 0 | 0 | 0 |
| 2012* | 0 | 0 | 0 | 0 | 0 | 0 |
| 2011 | 0 | 0 | 1 | 0 | 0 | 0 |
| 2010 | 0 | 1 | 0 | 0 | 0 | 0 |
| 2009* | 0 | 0 | 0 | 0 | 0 | 0 |
| 2008 | 1 | 0 | 0 | 0 | 0 | 0 |
| 2007 | 0 | 1 | 0 | 0 | 0 | 0 |
| 2006 | 1 | 0 | 0 | 0 | 0 | 0 |
| total | 6 | 4 | 2 | 1 | 2 | 5 |

(guerrilla warfare strategy). Our model thus indicates that China adopts a weaker player strategy, which seems to contradict that it has a stronger military force and a better developed border infrastructure [56]. However, it has been observed that China's strength in numbers could be misleading and India's military position may be stronger than usually acknowledged by its military analysts [12, 57]. The Indian army has established new mountain divisions and improved its access to the red-zones, after the completion of the all weather Darbuk–Shyok–DBO Road in 2019. Indeed, this may have spurred an increase of the Chinese activities [24, 58].

Game theory predicts that the number of incursions per red-zone averages out over time [52]. If we combine the relatively closely located Galwan valley and the Hotspring area into one, then the distribution of the incidents over the resulting five red-zones is (0.23, 0.11, 0.26, 0.16, 0.24), while game theory predicts a distribution of 20% per red-zone. A chi-square test of the hypothesis that the incursions are drawn from a uniform distribution is inconclusive with a $p$-value of 0.09. The red-zones Galwan and Hotspring are relatively small and if we remove them altogether, then the $p$-value is 0.58. This indicates that the Chinese incursions in the Aksai Chin average out over time and that the four major red-zones in that area are of equal

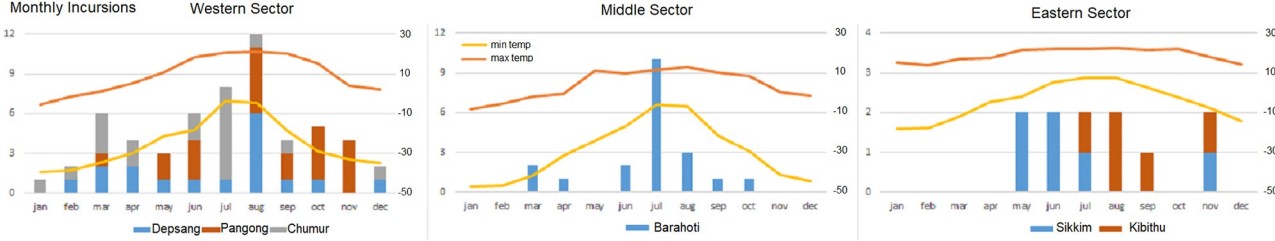

**Fig 8. Incursions per month versus temperature.** Incursions per month in the three sectors (for selected red-zones) show that the majority occur in summer. In the western sector, the incursions in Pangong Lake occur mainly during August-November while those in Chumur occur mainly in January-July. The weather in the eastern sector is less extreme than in the western/middle sector.

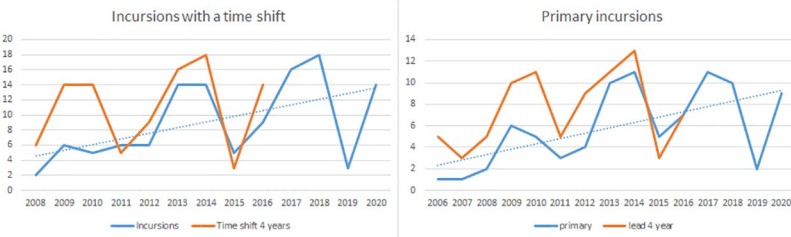

**Fig 9. The periodicity of the conflict.** The number of incursions under a time shift of 4 years. The overlapping graphs illustrate the periodicity of the conflict (auto correlation 0.73). The graph of primary incursions produce a similar overlap under a lead of 4 years (auto correlation 0.75) and upward trend.

strategic importance to the PLA. Recent developments support this. In our data, the number of incursions in Demchok is below average, but the PLA is building up its presence there. In December 2020, it constructed a new housing colony in this red-zone [59]. Similar constructions have been observed all along the LAC, thanks to satellite imagery of Maxar Technologies [32, 57]. We conclude that a Blotto game models the military strategy of the PLA and the incursions are strategically planned to gain control over the red-zones.

In the eastern sector, the conflict is much more controlled, see Table 6, although the number of incursions increased markedly in 2016, which is the year before the Doklam standoff. There has been a remarkable rise in number of incursions in Kibithu, at the easternmost part of the border, close to Myanmar. The reason behind this is unclear, as this is not a region of strategic or economic importance. In the eastern sector, the number of incursions per red-zone is too low to draw any conclusions that are statistically meaningful.

The Chinese incursions in both the western and the eastern sector target a few red-zones at a time. In the western sector we can model the incursions by a Blotto game and our data matches the randomized allocation which is optimal in such a game. The incursions are clustered in time and space, but average out over time. This indicates that there is an ongoing competition for control of all the red-zones, in which the PLA has constructed semi-permanent fortifications. This may, however, not be a sign of strength as the strategy of the PLA is similar to that of a weaker player in a Blotto game.

## Incursions over time

The majority of the incursions take place during the summer season. This is demonstrated in Fig 8, where we present the number of monthly incursions and extremal temperatures (observed min and max during 2007–2021). The red-zones are at high altitudes of over 4,000m and temperatures are extremely low in winter. The majority of the incursions take place at moderate temperatures of 5˚C or above. The exception is Depsang, where 40% of the incursions took place at extreme temperatures of around −20˚. Incursions at such temperatures have occurred in other red-zones, but only rarely. Due to the seasonality of the incursions and the modest size of our data set, we mainly consider the number of incursions per year.

The 21 day Depsang standoff in April 2013 was a decisive moment in the border dispute. It was followed by a sharp increase in incursions and a sequence of standoffs and skirmishes (Chumur 2014, 16 days; 2015 Burtse, 5 days; Doklam 2017, 73 days; Galwan 2020, 2 days). As a result of this, we cannot expect the time-series of incursions to be stationary. However, it clearly shows periodic fluctuations around an upward trend, as illustrated in Fig 9. The primary incursions show the same 4 year periodicity and the same upward trend. The periodic

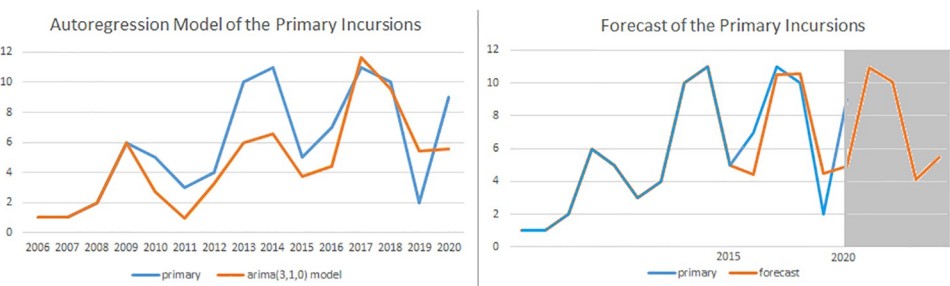

**Fig 10. Autoregression of the primary incursions.** We applied an ARIMA(3,1,0) model to the time-series (lag 3, integration 1, moving average 0) and used it to forecast the period 2015–2024. It predicts that the cyclicity of the conflict will persist.

**Table 7. Correlation between the primary incursions in the west and the east.** We combine the western and eastern sector in one sector and apply varying leads between -3 and 3 years. A positive lead (predicting the incursions in the east) clearly increases the correlation. A negative lead (predicting the incursions in the west) has no significant effect.

| lead (years) | -3 | -2 | -1 | 0. | 1 | 2 | 3 |
|---|---|---|---|---|---|---|---|
| corr. coef. | -0.10 | 0.04 | 0.17 | -0.02 | -0.02 | 0.40 | 0.74 |

fluctuation around the trend is now also clearly visible in the incursions prior to 2013. According to the KPSS test (3 lags) the primary incursions are trend stationary (p-value 0.03).

We applied an ARIMA(3,1,0) model to the time-series of primary incursions as illustrated in Fig 10. Note that the model underpredicts the increase of 2013, which is the year of the Depsang standoff. It overpredicts the number of incursions in 2019, when the Indian media underreported Chinese incursions. The conflict is dominated by sudden military incidents or political developments, which are hard to pick up in an autoregression. However, the model shows the same periodicity as the data, but its fluctuation is more moderate. A forecast of the next four years predicts that this fluctuation persists.

The number of primary incursions in the western/middle sector and the eastern sector is uncorrelated (-0.02) but varies under a time shift, as shown in Table 7. As is to be expected, the maximum correlation occurs for a lead of 3 years, because of the periodicity of the time-series. In particular, a positive lead of 3 years for incursions in the west gives a high correlation (0.74). Again, this is as expected. The Depsang standoff, which is the first major incident in the west, preceded the Doklam standoff in the east by four years. In our analysis of incursions per sector, we found that these are independent events. However, if we compare the incursions under a time shift, then we observe that the tension of the conflict fluctuates in sync along the entire LAC.

## Conclusion

We assembled a data set on incursions along the LAC that were reported in the media. From our analysis we conclude that Chinese incursions in the west and in the east are independent. Militarily, the west and east can be seen as two different conflicts. Furthermore, the Chinese incursions do not seem to be random encounters, but are strategically planned in line with the optimal play in a Blotto game. However, the fluctuation of the tension (the number of yearly incursions) appears to be in sync in the west and the east. Tensions rise after major skirmishes or standoffs, which occur in the most contested red-zones of Depsang, Pangong, Chumur, Barahoti, and Doklam. Such standoffs are followed by bilateral talks to avoid further escalation of

the conflict. The two countries remain at a constant state of high alertness. There are no signs that this situation will improve in the near future.

Our data set was built from media reports. In order to deal with the self-reinforcing nature of media attention, we considered incursions that were not preceded by another incursion in the same red-zone. This adjusts for over reporting by the media. It does not adjust for under reporting, which happened in the year 2019. The fluctuation that we find in our analysis may be less strong in reality. The ability of the political leaders to downplay media attention could be an advantage for reaching an agreement. It could also be a disadvantage, if the media attention is drummed up. The future development of the conflict is in the hands of the political leaders.

China's foreign policy has become increasingly aggressive, stepping up its military exercises around Taiwan and extending its presence in the South China Sea. To counter China's expansive policies, Australia, the UK, and the USA have entered a partnership and one option for India is to align itself with the AUKUS countries. The military effort to step up its presence in the red-zones and counter the Chinese incursions, requires an enormous effort by India that can only be achieved through assistance by a strong partnership. This appears to be the approach that is favored by the Modi administration. It has strengthened its ties with Australia, Japan, and the USA ("the Quad") to counter Chinese expansion in the Indo-Pacific region.

Another option for India is a continued diplomatic approach, through ongoing bilateral talks with China. This may seem like a dead-end, since diplomacy has not made much progress over sixty years. On the contrary, Modi's efforts of rapprochements with China have failed, and India imposed economic sanctions against China after the Galwan incident. The best that bilateral talks have achieved so far is defusing conflicts to prevent further escalation. In spite of all this, there are several arguments that can be made in favor of a renewed diplomatic approach. The two countries are inextricably linked economically. A reduced military presence would reduce the ecological footprint. According to our analysis, only a few contested areas lie at the root of the conflict, and seem to generate incursions in other red-zones all over the LAC. The two countries could try to reach an agreement on only a limited part of the LAC. A good starting point would be an agreement on the border in the Sikkim sector and its nearby disputed zones on the border between China and Nepal or Bhutan. This would defuse the conflict in the eastern sector, and if our analysis is right, likely also in the western sector. This could be an important first step in a step-by-step resolution of the entire conflict.

Unfortunately, recent developments indicate that China is taking steps in the wrong direction. China and Bhutan have just signed a bilateral agreement to settle their dispute about the Doklam plateau, leaving India out of the equation. The Chinese media have presented this agreement as a snub to the Indian government [60], which does not bode well. If China pursues this approach, this could lead to a worsening of the conflict, leaving a strong military alliance with the AUKUS countires as the only option for India.

## Supporting information

**S1 Text.**
(TXT)

## Author Contributions

**Conceptualization:** V. S. Subrahmanian.

**Data curation:** Jan-Tino Brethouwer, Robbert Fokkink, Kevin Greene, Roy Lindelauf, Caroline Tornquist.

**Formal analysis:** Jan-Tino Brethouwer, Robbert Fokkink, Kevin Greene, Roy Lindelauf, Caroline Tornquist, V. S. Subrahmanian.

**Writing – original draft:** Jan-Tino Brethouwer, Robbert Fokkink.

**Writing – review & editing:** Jan-Tino Brethouwer, Robbert Fokkink, Kevin Greene, Roy Lindelauf, V. S. Subrahmanian.

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
