## [Decision Letter · Decision Letter 0]

11 Apr 2022

PONE-D-22-04496Rising tension in the Himalayas: a geospatial analysis of Chinese border incursions into IndiaPLOS ONE

Dear Dr. Fokkink,

Thank you for submitting your manuscript to PLOS ONE. After careful consideration, we feel that it has merit but does not fully meet PLOS ONE’s publication criteria as it currently stands. Therefore, we invite you to submit a revised version of the manuscript that addresses the points raised during the review process.

We look forward to receiving your revised manuscript.

Kind regards,

Manoj Kumar

Academic Editor

PLOS ONE

Journal Requirements:

5. Please ensure that you refer to Figures 1 in your text as, if accepted, production will need this reference to link the reader to the figure.

6. We note that Figures 1, 3, and 5 in your submission contain map images which may be copyrighted. All PLOS content is published under the Creative Commons Attribution License (CC BY 4.0), which means that the manuscript, images, and Supporting Information files will be freely available online, and any third party is permitted to access, download, copy, distribute, and use these materials in any way, even commercially, with proper attribution. For these reasons, we cannot publish previously copyrighted maps or satellite images created using proprietary data, such as Google software (Google Maps, Street View, and Earth). For more information, see our copyright guidelines: http://journals.plos.org/plosone/s/licenses-and-copyright.

a. You may seek permission from the original copyright holder of Figures 1, 3, and 5 to publish the content specifically under the CC BY 4.0 license.  

Reviewers' comments:

Reviewer's Responses to Questions

**Comments to the Author**

1. Is the manuscript technically sound, and do the data support the conclusions?

Reviewer #1: Yes

Reviewer #2: Partly

2. Has the statistical analysis been performed appropriately and rigorously? 

Reviewer #1: Yes

Reviewer #2: No

3. Have the authors made all data underlying the findings in their manuscript fully available?

Reviewer #1: No

Reviewer #2: Yes

4. Is the manuscript presented in an intelligible fashion and written in standard English?

Reviewer #1: Yes

Reviewer #2: Yes

5. Review Comments to the Author

Reviewer #1: The manuscript has attempted to understand the spatial dynamics of the Chinese border incursions into India along the Himalayas by employing statistical methods and game theory. The paper has done a satisfactory job, and maybe accepted for publication. A few minor revisions are suggested. The suggestions are attached along with this.

Reviewer #2: PONE-D-22-04496

“Rising tension in the Himalayas: a geospatial analysis of Chinese border incursions into India”

Summary

This manuscript aims to build an understanding of what drives the conflict between India and China in their contested border regions. The authors have assembled an original data set on Chinese incursions into India over the period 2006-2020. They apply statistical and game-theoretic methods to study the dynamic of rising and falling tensions along the LAC. Their findings indicate a possible way out of the conflict through a step-by-step de-escalation.

Evaluation:

The analysis makes some interesting points and the data is new, but there are a number of issues. It is not clear how this data relates to other standard conflict datasets, the theoretical treatment of the literature on Blotto games and systems defense does not accurately represent what theory tell us about incentives related to this application, and the statistical analysis is not what would be considered the current standard in empirical conflict studies. It is possible all of these issue could be addressed in revisions, though I think because there are so few observations in the data it will be difficult to make many conditional statements about the effect of the variables of interest on incursion strategies they want to claim. But obviously that is an empirical question.

I am also a little skeptical that we can conclude that the Chinese incursions in the west are strategically planned and aim for a permanent control, or at least a clear status quo of the contested areas from this analysis. I think we can conclude that maybe that is true, maybe asymmetric valuations are the cause, or asymmetric budget constraints. We can conclude that the targeting is not consistent with the incentives of a symmetric zero-sum military competition for territory.

Other Comments:

p.2 line 19 Surplus in what?

Is this data comparable to say the widely used ACLED database on conflict events?

What are the statistical implications for the selections of observations, i.e., China only and only those verified by multiple sources? What happens to the results if you include the unverified reports or Indian incursions?

In the symmetric Blotto game uniform randomization is the equilibrium strategy, with asymmetric values and resources, the marginal distributions are more complicated. It is not clear to me how these tests speak to the equilibrium of that game under the conditions of this problem.

Doesn’t local geography and weather matter for the rates of incursions in the west and east? I was expecting some sort of conditional analysis (i.e., with controls related to target location). The basic thought experiment I have in mind is: suppose that China wanted to attack in the east and west every day, but can only do so when the weather is good. The daily weather is random and uncorrelated at the two locations, then the unconditional observation of incursions would be uncorrelated, even though, in fact, they are the product of the same perfectly correlated strategy: attack everyday.

The description of the data is not as clear as it probably should be.

I don’t understand this claim: “The border dispute can therefore be divided into two separate conflicts, centered around Aksai Chin and Arunachal Pradesh.” The whole point of the NE in the Blotto game is that the conflicts are connected, but the strategic interaction (at least in the symmetric case) induces strategies that are uniform randomizations.

When we got the periodicity analysis, I was expecting some kind of time series statistics, lagged regression analysis, or something. It is not clear to me exactly what the authors did here (It looks like simply doing correlations between y_t and y_{t-k} for some k’s), but it is not what I would consider standard practice to analyze time dependence in panel data.

6. PLOS authors have the option to publish the peer review history of their article (what does this mean?). If published, this will include your full peer review and any attached files.

Reviewer #1: No

Reviewer #2: No

---

## [Author Response · Author response to Decision Letter 0]

8 Jul 2022

We have uploaded a "response to reviewers" that addresses the reviewer comments and describes how we have revised the paper.

---

## [Decision Letter · Decision Letter 1]

9 Sep 2022

Rising tension in the Himalayas: a geospatial analysis of Chinese border incursions into India

PONE-D-22-04496R1

Dear Dr. Fokkink,

We’re pleased to inform you that your manuscript has been judged scientifically suitable for publication and will be formally accepted for publication once it meets all outstanding technical requirements.

Kind regards,

Manoj Kumar

Academic Editor

PLOS ONE

Additional Editor Comments (optional):

Reviewers' comments:

Reviewer's Responses to Questions

**Comments to the Author**

1. If the authors have adequately addressed your comments raised in a previous round of review and you feel that this manuscript is now acceptable for publication, you may indicate that here to bypass the “Comments to the Author” section, enter your conflict of interest statement in the “Confidential to Editor” section, and submit your "Accept" recommendation.

Reviewer #1: (No Response)

Reviewer #2: All comments have been addressed

2. Is the manuscript technically sound, and do the data support the conclusions?

Reviewer #1: Yes

Reviewer #2: Yes

3. Has the statistical analysis been performed appropriately and rigorously? 

Reviewer #1: Yes

Reviewer #2: Yes

4. Have the authors made all data underlying the findings in their manuscript fully available?

Reviewer #1: Yes

Reviewer #2: Yes

5. Is the manuscript presented in an intelligible fashion and written in standard English?

Reviewer #1: Yes

Reviewer #2: Yes

6. Review Comments to the Author

Reviewer #1: (No Response)

Reviewer #2: (No Response)

7. PLOS authors have the option to publish the peer review history of their article (what does this mean?). If published, this will include your full peer review and any attached files.

Reviewer #1: No

Reviewer #2: No

---

## [Editor Report · Acceptance letter]

14 Sep 2022

PONE-D-22-04496R1 

Rising tension in the Himalayas: a geospatial analysis of Chinese border incursions into India 

Dear Dr. Fokkink:

I'm pleased to inform you that your manuscript has been deemed suitable for publication in PLOS ONE. Congratulations! Your manuscript is now with our production department. 

Kind regards, 

on behalf of

Dr. Manoj Kumar 

Academic Editor

PLOS ONE